# Invertible Gaussian Reparameterization: Revisiting the Gumbel-Softmax

**Andres Potapczynski**
Zuckerman Institute
Columbia University
ap3635@columbia.edu

**Gabriel Loaiza-Ganem**
Layer 6 AI*
gabriel@layer6.ai

**John P. Cunningham**
Department of Statistics
Columbia University
jpc2181@columbia.edu

## Abstract

The Gumbel-Softmax is a continuous distribution over the simplex that is often used as a relaxation of discrete distributions. Because it can be readily interpreted and easily reparameterized, it enjoys widespread use. We propose a modular and more flexible family of reparameterizable distributions where Gaussian noise is transformed into a one-hot approximation through an invertible function. This invertible function is composed of a modified softmax and can incorporate diverse transformations that serve different specific purposes. For example, the stick-breaking procedure allows us to extend the reparameterization trick to distributions with countably infinite support, thus enabling the use of our distribution along nonparametric models, or normalizing flows let us increase the flexibility of the distribution. Our construction enjoys theoretical advantages over the Gumbel-Softmax, such as closed form $\mathbb{KL}$, and significantly outperforms it in a variety of experiments. Our code is available at `https://github.com/cunningham-lab/igr`.

## 1 Introduction

Numerous machine learning tasks involve optimization problems over discrete stochastic components whose parameters we wish to learn. Mixture and mixed-membership models, variational autoencoders, language models and reinforcement learning fall into this category [13, 14, 26, 17, 8]. Ideally, as with fully continuous models, we would use stochastic optimization via backpropagation. One strategy to compute the necessary gradients is using score estimators [8, 32], however these estimates suffer from high variance which leads to slow convergence. Another strategy is to find a reparameterizable continuous relaxation of the discrete distribution. Reparameterization gradients exhibit lower variance but are contingent on finding such a relaxation. Jang et al. [12] and Maddison et al. [19] independently found such a continuous relaxation via the Gumbel-Softmax (GS) or Concrete distribution.

The GS has experienced wide use and has been extended to other settings, such as permutations [18], subsets [33] and more [1]. Its success relies on several qualities that make it appealing: $(i)$ it is reparameterizable, that is, it can be sampled by transforming parameter-independent noise through a smooth function, $(ii)$ it can approximate any discrete distribution, (i.e. converge in distribution) $(iii)$ it has a closed form density, and $(iv)$ its parameters can be interpreted as the discrete distribution that it is relaxing. While the last quality is mathematically pleasing, it is not a necessary condition for a valid relaxation. Here we ask: *how important is this parameter interpretability*? In the context of deep learning models, interpreting the parameters is not a first concern, and we show that the GS can be significantly improved upon by giving up this quality.

In this paper we propose an alternative family of distributions over the simplex to achieve this relaxation, which we call Invertible Gaussian Reparameterization (IGR). Our reparameterization

works by transforming Gaussian noise through an invertible transformation onto the simplex, and a temperature hyperparameter allows the distribution to concentrate its mass around the vertices. IGR is more natural, more flexible, and more easily extended than the GS. Furthermore, IGR enables using the reparameterization trick on distributions with countably infinite support, which enables nonparametric uses, and also admits closed form $\mathbb{KL}$ divergence evaluation. Finally, we show that our distribution outperforms the GS in a wide variety of experimental settings.

## 2  Background

### 2.1  The reparameterization trick

Many problems in machine learning can be formulated as optimizing parameters of a distribution over an expectation:

$$\phi^* = \arg\max_{\phi} L(\phi) := \arg\max_{\phi} \mathbb{E}_{q_\phi(z)}[f(z)] \tag{1}$$

where $q_\phi$ is a distribution over $\mathcal{S}$ parameterized by $\phi$ and $f : \mathcal{S} \to \mathbb{R}$. In order to use stochastic gradient methods [27, 3], the gradient of $L$ has to be estimated. A first option is to use score estimators [8, 32]. However, in practice score estimators usually exhibit high variance [20]. The reparameterization trick [14] provides an alternative estimate of this gradient which empirically has less variance, resulting in more efficient optimization. The reparameterization trick consists of finding a function $g(\epsilon, \phi)$ such that $g$ is differentiable with respect to $\phi$ and if $z \sim q_\phi$, then:

$$z \overset{\mathrm{d}}{=} g(\epsilon, \phi) \tag{2}$$

where $\epsilon$ is a continuous random variable whose distribution does not depend on $\phi$ and is easy to sample from. The gradient is then estimated by:

$$\nabla_\phi L(\phi) \approx \frac{1}{B} \sum_{b=1}^{B} \nabla_\phi f(g(\epsilon_b, \phi)) \tag{3}$$

where $\epsilon_1, \ldots, \epsilon_B$ are iid samples from the distribution of $\epsilon$. For example, if $\phi = (\mu, \sigma)$ and $q_\phi = \mathcal{N}(\mu, \sigma^2)$ then the reparameterization trick is given by $g(\epsilon, \phi) = \mu + \sigma\epsilon$ with $\epsilon \sim \mathcal{N}(0, 1)$.

### 2.2  Continuous relaxations

While we can use score estimators whether $q_\phi$ has continuous or discrete support, the reparameterization gradient of equation 3 is only valid when $q_\phi$ has continuous support. To extend the reparameterization trick to the discrete setting, thus avoiding the high variance issues of score estimators, suppose $q_\phi$ is a distribution over the set $\mathcal{S} = \{1, 2, \ldots, K\}$. We use one-hot representations of length $K$ for the elements of $\mathcal{S}$, so that $\mathcal{S}$ can be interpreted as the vertices of the $(K-1)$-simplex, $\Delta^{(K-1)} = \{z \in \mathbb{R}^K : z_k \geq 0 \text{ and } \sum_{k=1}^{K} z_k = 1\}$. The idea is to now place a continuous distribution over $\Delta^{(K-1)}$ that approximates $q_\phi$. Note that placing a distribution over $\Delta^{(K-1)}$ is equivalent to placing a distribution over $\mathcal{S}^{(K-1)} = \{z \in \mathbb{R}^{K-1} : z_k > 0 \text{ and } \sum_{k=1}^{K-1} z_k < 1\}$, as the last coordinate can be obtained from the previous ones: $z_K = 1 - \sum_{k=1}^{K-1} z_k$. Placing a distribution over $\mathcal{S}^{(K-1)}$ is mathematically convenient as $\mathcal{S}^{(K-1)} \subset \mathbb{R}^{K-1}$ has positive Lebesgue measure, while $\Delta^{(K-1)} \subset \mathbb{R}^K$ does not. Although this distinction might appear as an irrelevant technicality, it allows us to correctly compute our Jacobians in section 3. We will thus interchangeably think of distributions over $\mathcal{S}$ as points in either $\mathcal{S}^{(K-1)}$ or $\Delta^{(K-1)}$. The optimization problem of equation 1 is then relaxed to:

$$\tilde{\phi}^* = \arg\max_{\tilde{\phi}} \tilde{L}(\tilde{\phi}) := \arg\max_{\tilde{\phi}} \mathbb{E}_{\tilde{q}_{\tilde{\phi}}(\tilde{z})}[\tilde{f}(\tilde{z})] \tag{4}$$

where $\tilde{q}_{\tilde{\phi}}$ is a distribution over $\mathcal{S}^{(K-1)}$ and the function $\tilde{f} : \mathcal{S}^{(K-1)} \to \mathbb{R}$ is a relaxation of $f$ to $\mathcal{S}^{(K-1)}$. As long as $\tilde{q}_{\tilde{\phi}}$ concentrates most of its mass around $\mathcal{S}$ and $\tilde{f}$ is smooth, this relaxation is sensible. If $\tilde{q}_{\tilde{\phi}}$ can be reparameterized as in equation 2, then we can use the reparameterization trick. We make two important notes: first, not only the distribution is relaxed, the function $f$ also has to

be relaxed to $\tilde{f}$ because it now needs to take inputs in $\mathcal{S}^{(K-1)}$ and not just $\mathcal{S}$. In other words, the objective must also be relaxed, not just the distribution. Second, the parameters $\tilde{\phi}$ of the relaxed distribution need not match the parameters $\phi$ of the original distribution.

Maddison et al. [19] and Jang et al. [12] proposed the Gumbel-Softmax distribution, which is parameterized by $\alpha \in (0, \infty)^K$ and a temperature hyperparameter $\tau > 0$, and is reparameterized as:

$$\tilde{z} \stackrel{\mathsf{d}}{=} \text{softmax}\big((\epsilon + \log \alpha)/\tau\big) \tag{5}$$

where $\epsilon \in \mathbb{R}^K$ is a vector with independent Gumbel$(0, 1)$ entries and $\log$ refers to elementwise logarithm. Note that when the temperature approaches $0$, not only does the GS concentrate its mass around $\mathcal{S}$, but it converges to a distribution proportional to $\alpha$. The GS distribution implied by equation 5 can be shown to be:

$$\tilde{q}_{\alpha,\tau}(\tilde{z}) = (K-1)! \; \tau^{K-1} \prod_{k=1}^{K} \left( \frac{\alpha_k \tilde{z}_k^{-\tau-1}}{\sum_{j=1}^{K} \alpha_j \tilde{z}_j^{-\tau}} \right) \tag{6}$$

We highlight the difference between $\alpha$ and $\tilde{\phi}$: the former is the parameter of the GS distribution and might depend on the latter, which is the parameter of the loss with respect to which we optimize in equation 4. For example, $\alpha$ might be the output of a neural network parameterized by $\tilde{\phi}$. A common use of the GS is to relax objectives of the form:

$$\mathbb{KL}(q_\phi || p_0) = \mathbb{E}_{q_\phi(z)} \left[ \log \frac{q_\phi(z)}{p_0(z)} \right] \tag{7}$$

where $p_0$ is a distribution over $\mathcal{S}$. Relaxing this $\mathbb{KL}$ requires additional care: it cannot be relaxed to $\mathbb{KL}(\tilde{q}_{\tilde{\phi}} || p_0)$ because the $\mathbb{KL}$ divergence is not well defined between a continuous and a discrete distribution. In other words, relaxing $f$ to $\tilde{f}$ is not straightforward when a $\mathbb{KL}$ divergence is involved in the objective. When using a GS relaxation, researchers commonly replace this $\mathbb{KL}$ with [12, 6, 31]:

$$\mathbb{KL}(\bar{\alpha} || p_0) \text{ where } \bar{\alpha}_k = \frac{\alpha_k}{\sum_{i=1}^{K} \alpha_i} \tag{8}$$

the idea being that, for low temperatures, the GS approximates a distribution proportional to its parameter, i.e. $\bar{\alpha} \in \Delta^{(K-1)}$. The goal of this substitution is to still compute a $\mathbb{KL}$ between two discrete variables, even after relaxing the distribution. This substitution is problematic, as pointed out by Maddison et al. [19], as it does not take into account how close the GS actually is to $\bar{\alpha}$. A more sensible way to relax the discrete $\mathbb{KL}$ is to relax it to an actual continuous $\mathbb{KL}$ as done by Maddison et al. [19]:

$$\mathbb{KL}(\tilde{q}_{\tilde{\phi}} || \tilde{q}_0) \tag{9}$$

where $\tilde{q}_0$ is fixed in such a way that it is close to $p_0$. For the GS, finding such a distribution is straightforward as a consequence of its parameter interpretability: $\tilde{q}_0$ can be chosen as a GS with parameter $\alpha_0 = p_0$. Note that the $\mathbb{KL}$ in equation 9 cannot be directly evaluated, but a Monte Carlo estimate can be formed thanks to the closed form density of equation 6 and thus stochastic gradient descent can be performed.

Finally, it is worth remarking that while Stirn et al. [30] and Gordon-Rodriguez et al. [9] proposed distributions over the simplex which admit reparameterization gradients, their goals are not to obtain discrete relaxations. Thus they do not have a temperature hyperparameter allowing to concentrate mass on the vertices to approximate discrete distributions.

## 3   The invertible Gaussian reparameterization family

If the only requirements for a continuous relaxation are a reparameterizable distribution on the simplex and a temperature hyperparameter allowing to concentrate mass around the vertices, one might logically ask: why use the specific choices of the GS? Namely, why use the unusual Gumbel noise and be forced to use the softmax as a mapping onto the simplex? If tasked with constructing a reparametrizable distribution on the simplex, we argue that the most natural choice is to sample Gaussian noise and map it to the simplex; trying different mappings and keeping the best performing one. The cost of this choice is losing the parameter interpretability of the GS, but we will show the advantages are numerous and well worth this cost.

We now present the IGR distribution on $\mathcal{S}^{(K-1)}$, which is parameterized by $(\mu, \sigma)$, where $\mu \in \mathbb{R}^{K-1}$ and $\sigma \in (0, \infty)^{K-1}$. Gaussian noise $\epsilon = (\epsilon_1, \ldots, \epsilon_{K-1}) \sim \mathcal{N}(0, I_{K-1})$ is transformed in the following way:

$$y = \mu + \text{diag}(\sigma)\epsilon \tag{10}$$
$$\tilde{z} = g(y, \tau) \tag{11}$$

where $\text{diag}(\sigma)$ is a diagonal matrix whose nonzero elements are given by $\sigma$, $g(\cdot, \tau)$ is an invertible smooth function and $\tau > 0$ is a temperature hyperparameter. Note that IGR is not only more natural than the GS, but is is also more flexible, having $2K - 2$ parameters instead of $K$. The first advantage of choosing $g$ to be an invertible function is that the density of $\tilde{z}$ can be computed in closed form with the change of variable formula:

$$\tilde{q}_{\mu, \sigma, \tau}(\tilde{z}) = \mathcal{N}(y|\mu, \sigma)|\det J_g(y, \tau)|^{-1} \tag{12}$$

where $J_g(\cdot, \tau)$ is the Jacobian of $g(\cdot, \tau)$. The second advantage of this choice is that it allows us to compute the $\mathbb{KL}$ in closed form (as the Jacobian terms cancel out in the ratio):

$$\mathbb{KL}\left(\tilde{q}_{\mu, \sigma, \tau}(\tilde{z})||\tilde{q}_{\mu_0, \sigma_0, \tau}(\tilde{z})\right) = \mathbb{KL}\left(\mathcal{N}(\mu, \sigma^2)||\mathcal{N}(\mu_0, \sigma_0^2)\right) \tag{13}$$

and thus Monte Carlo estimation of equation 9 is no longer needed.

The components of the IGR can be easily mixed-and-matched. For example, while we use Gaussian noise as the most natural first choice because it is reparameterizable and because the $\mathbb{KL}$ divergence between two Gaussians has closed form, any other choice with these two properties can also be used. Similarly, any choice of $g$, as long as it obeys some requirements which we explain in the section 3.1, can also be used. In contrast, changing the Gumbel distribution or the softmax used in the GS cannot be done. These properties of the IGR make it more easily extensible than the GS.

Since the parameter interpretability of the GS is lost in IGR, we cannot directly read $\mu_0$ and $\sigma_0$ from $p_0 \in \mathcal{S}^{(K-1)}$. Thus when a $\mathbb{KL}$ term is involved, while IGR gains the ability to evaluate it analytically, we solve the following optimization problem to obtain these parameters:

$$(\mu_0, \sigma_0) = \underset{(\mu, \sigma)}{\arg\min} \, \mathbb{E}_{\tilde{q}_{\mu, \sigma, \tau}(\tilde{z})}[||\tilde{z} - p_0||_2^2] \tag{14}$$

Note that having to solve this problem is a very small price to pay for losing parameter interpretability: the optimization is a very simple moment matching problem and has to be be computed only once for any given $p_0$.

## 3.1   Choosing $g(\cdot, \tau)$

In this section we design some invertible functions that could be used and argue the rationale behind their construction. There are two important desiderata for $g(\cdot, \tau)$: the first one is that we should be able to compute the determinant of its Jacobian efficiently, which enables tractable density evaluation. This tractability can be achieved, for example, by ensuring the Jacobian is triangular. Note that although in many instances we do not actually require evaluating the density of the relaxation (e.g. variational autoencoders [14]), this is a problem-specific property and density evaluation remains desirable in general. The second is that the limit as $\tau \to 0$ of $g(y, \tau)$ is in $\mathcal{S}$ for almost all $y$, meaning that as the temperature gets smaller, the distribution places most of its mass around the vertices. The two most natural choices for mapping onto the simplex are the softmax function and the stick-breaking procedure. As we explain below, these alone are not enough, and we thus modify them to make them appropriate for our purposes. The softmax has two issues: first, it maps to $\Delta^{(K-1)}$ and not $\mathcal{S}^{(K-1)}$ and second, it is not invertible. Both of these problems can be addressed with a small modification of the softmax function:

$$g(y, \tau)_k = \frac{\exp(y_k/\tau)}{\sum_{j=1}^{K-1} \exp(y_j/\tau)) + \delta} \tag{15}$$

where $\delta > 0$ ensures that the function is invertible and maps to $\mathcal{S}^{(K-1)}$. Furthermore, the Jacobian of this transformation can be efficiently computed with the matrix determinant lemma (see appendix for details). We will refer to this transformation as the softmax$_{++}$.

The other natural alternative to map from $(0,1)^{K-1}$ onto $\mathcal{S}^{(K-1)}$ is through the stick-breaking procedure [7], which we briefly review here. Given $u \in (0,1)^{K-1}$, the result $v = SB(u)$ of performing stick-breaking on $u$ is given by:

$$v_k = u_k \prod_{i=1}^{k-1} (1 - u_i), \text{ for } k = 1, 2, \ldots, K - 1 \tag{16}$$

In addition to producing outputs in $\mathcal{S}^{(K-1)}$, this procedure has some useful properties, namely: it is invertible, its Jacobian is triangular, and it can easily be extended to the case where $K = \infty$ (which will be useful to extend IGR to relax discrete distributions with countably infinite support). While the invertibility property might suggest that the stick-breaking procedure alone is enough to use with IGR, a temperature hyperparameter $\tau$ still needs to be introduced in such a way that as $\tau \to 0$, the resulting distribution concentrates its mass on the vertices. Unlike with the softmax$_{++}$, simply dividing the input by $\tau$ does not achieve this limiting behavior. The most natural way of introducing a temperature that achieves the desired limiting behavior is by linearly interpolating to the nearest vertex, resulting in a $g$ function given by:

$$\begin{cases} w = SB\big(\text{sigmoid}(y)\big) \\ g(y, \tau) = \tau w + (1 - \tau) P_{\mathcal{S}}(w) \end{cases} \tag{17}$$

where $P_{\mathcal{S}}$ is the projection onto the vertices of $\mathcal{S}^{(K-1)}$. Note that the Jacobian of this transformation is triangular. However, we found better empirical performance with the following function, which introduces the temperature using the softmax$_{++}$ function:

$$g(y, \tau) = \text{softmax}_{++}(w, \tau) \tag{18}$$

While it might seem redundant to apply both a stick-breaking procedure and a softmax$_{++}$ as they both map to $\mathcal{S}^{(K-1)}$, the softmax$_{++}$ function allows to introduce $\tau$ in such a way that the distribution concentrates its mass around the vertices as $\tau \to 0$. Also, as seen in section 3.2, the stick-breaking procedure proves useful as it enables using the reparameterization trick in the countably infinite support setting.

Finally, another choice of $g(\cdot, \tau)$ could be a normalizing flow [25, 15, 5] followed by softmax$_{++}$. Normalizing flows are flexible neural networks constructed in such a way that they are invertible while still allowing tractable Jacobian determinant evaluations, so that they enable us to learn $g$. We note that normalizing flows require additional parameters, so that when using them, IGR is not only parameterized by $\mu$ and $\sigma$, but by the parameters of the normalizing flow as well. Thus, if a $\mathbb{KL}$ is involved, the optimization problem of equation 14 needs to be solved over the parameters of the normalizing flow too, and as a result the $\mathbb{KL}$ in equation 9 cannot be evaluated in closed form anymore, as the parameters of the two involved normalizing flows need not match. However, Monte Carlo estimates of the $\mathbb{KL}$ are still readily available.

### 3.2 Reparameterization trick for countably infinite distributions

Since the stick-breaking procedure can map to $\mathcal{S}^{\infty} = \big\{ z \in \mathbb{R}^{\infty} : z_k \geq 0 \text{ and } \sum_{k=1}^{\infty} z_k = 1 \big\}$, we can extend equation 18 to the setting where the discrete distribution has countably infinite support (e.g. Poisson, geometric or negative binomial distributions). In this setting, the IGR is parameterized by $\mu \in \mathbb{R}^{\infty}$ and $\sigma \in (0, \infty)^{\infty}$. Clearly backpropagating through infinitely many parameters cannot be done in a computer, but we do not have to do so as most of the parameters contribute very little to the loss. For a sample $\epsilon_1, \epsilon_2, \ldots$ we only update the first $K$ coordinates of $\mu$ and $\sigma$, where $K$ is the number such that:

$$\sum_{k=1}^{K-1} g(y, \tau)_k \leq \rho < \sum_{k=1}^{K} g(y, \tau)_k \tag{19}$$

where $\rho \in (0,1)$ is a pre-specified precision hyperparameter and $g$ is as in equation 18. Note that here $K$ is now a random variable that depends on $\epsilon$ instead of being fixed as before, so that in a way the number of (effective) categories gets learned by the data. Note as well that the stick-breaking procedure is necessary to know where to cut $K$ as it guarantees that later terms in the sequence are small, which would not happen if we only applied a $\text{softmax}_{++}$ function.

### 3.3 Recovering the discrete distribution

Recall that the original objective of continuous relaxations is to solve the discrete problem of equation 1, so that once we have solved the continuous problem of equation 4, it is desirable to have the ability to recover a solution to the former problem. In other words, given the parameters of a continuous relaxation, we should be able to recover the discrete distribution that it is relaxing. The parameter intepretability of the GS allows to directly do so. In this section we derive a method for doing so with the IGR, which is enabled by the following proposition:

**Proposition 1**: For any $\delta > 0$, the following holds:

$$\lim_{\tau \to 0} \text{softmax}_{++}(y, \tau) = h(y) := \begin{cases} e_{k^*}, \text{ if } k^* = \underset{k=1,\ldots,K-1}{\arg\max} \ (y_k) \text{ and } \underset{k=1,\ldots,K-1}{\max} \ (y_k) > 0 \\ 0 \quad , \text{ if } \underset{k=1,\ldots,K-1}{\max} \ (y_k) < 0 \end{cases} \tag{20}$$

where $e_k \in \mathbb{R}^{K-1}$ is the one-hot vector with a 1 in its $k$-th coordinate.
**Proof**: See appendix.

Thus, the vector of discrete probabilities associated with IGR is $\mathbb{E}[h(\tilde{z})]$, which can be easily approximated through a Monte Carlo estimate by sampling from the IGR and averaging the results after transforming them with $h$. This is the last cost to pay for losing parameter interpretability, but once again it is very small: the complexity of this approximation is negligible when compared to the one of solving the problem of equation 4. Note also that this proposition enables the use of straight-through estimators [2], where the sample is discretized during the forward pass, but not for backpropagation. The next proposition shows that when just using the $\text{softmax}_{++}$ as $g$, the recovered discrete distribution can be written in an even more explicit form:

**Proposition 2**: If $y_k \sim \mathcal{N}(\mu_k, \sigma_k)$ for $k = 1, \ldots, K-1$, and we define the discrete random variable $H$ by $H = k$ if $h(y) = e_k$ and $H = K$ if $h(y) = 0$, then:

$$\mathbb{P}(H = k) = \begin{cases} \displaystyle\int_0^\infty \frac{1}{\sigma_k} \phi\left(\frac{t - \mu_k}{\sigma_k}\right) \prod_{j \neq k} \Phi\left(\frac{t - \mu_j}{\sigma_j}\right) dt, \text{ if } k = 1, \ldots, K-1 \\ \displaystyle\prod_{j=1}^{K-1} \Phi\left(-\frac{\mu_j}{\sigma_j}\right) \qquad\qquad\qquad , \text{ if } k = K \end{cases} \tag{21}$$

where $\phi$ and $\Phi$ are the standard Gaussian density and cumulative distribution function, respectively.
**Proof**: See appendix.

We finish this section by noting that there is literature proposing gradient estimators and attempting to reduce their variance [21, 20, 31, 10, 16]. In particular, Grathwohl et al. [10] and Tucker et al. [31] proposed techniques involving the GS. Their techniques, however, require computing the gradient of the discrete objective with respect to the parameters of the continuous relaxation, which can be done with the GS thanks to its parameter interpretability. Proposition 2 thus enables the use of their methods with IGR, as the integral in equation 21 can be easily approximated numerically. Due to space constraints we include details, along a discussion about bias, in the appendix.

## 4   Experiments

In this section, we contrast the performance of the IGR (with different choices of $g$) alongside that of the GS. First, in relation to section 3.2, we compare the ability of the IGR and the GS (with

varying number of categories) to approximate a countably infinite distribution. We then focus on tasks that involve a $\mathbb{KL}$ term in their objective function. Finally, we also consider a Structured Output Prediction task which does not involve a $\mathbb{KL}$ term. For the experiments involving a $\mathbb{KL}$ term, we use variational autoencoders (VAEs) [14]. We follow the setup of Maddison et al. [19] and Jang et al. [12] (although note that we use a slightly different objective than Jang et al. [12], see appendix for details). The datasets we use are handwritten digits from MNIST, fashion items from FMNIST and alphabet symbols from Omniglot. We ran each experiment 5 times and report averages plus/minus one standard deviation. Additionally, for all the experiments, we used the log scale implementation of the GS (ExpConcrete) as in Maddison et al. [19] since it avoids numerical issues and allows us to run the models involving the GS at lower temperatures. Throughout this section, the label IGR-I denotes the implementation with the softmax$_{++}$ (equation 15), the label IGR-SB the implementation with the stick-breaking transformation followed by a softmax$_{++}$ (equation 18), and finally the label IGR-Planar the implementation using two nested Planar Flows [25] followed by a softmax$_{++}$ .

Comparing any IGR variant against the GS requires selecting temperature hyperparameters for each model. To make a fair comparison, temperatures $\tau$ should be chosen carefully as they affect models differently, so they cannot just be set to the same value. We thus choose the temperature hyperparameter through cross validation, considering the range of possible temperatures $\{0.01, 0.03, 0.07, 0.1, 0.25, 0.4, 0.5, 0.67, 0.85, 1.0\}$ and compare best-performing models. However, and very importantly, we use the loss on the recovered *discrete* model — not the trained *continuous* one — to select the best performing model. This avoids the potential issue of having one model produce better discrete relaxations which are closer to the vertices of the simplex, while resulting in a larger continuous loss as the other model is allowed to use the simplex more freely. All implementation details are in the appendix.

### 4.1 Approximating a Poisson Distribution

Here we compare the ability of the IGR-SB and the GS to approximate distributions with countably infinite support. The top panels of Figure 1 show an approximation with the IGR-SB to a Poisson distribution with $\lambda = 50$, while the bottom panels show the same approximations when using a GS with different number $K$ of discrete components. These approximations are computed by optimizing the objective in equation 14. We can see how the IGR-SB outperforms the GS without having to specify $K$. We show further comparisons when approximating other distributions in the appendix.

### 4.2 Variational Autoencoders

We trained VAEs composed of 20 discrete variables with 10 categories each. VAEs are latent variable models which maximize the ELBO, a lower bound on the log likelihood involving a $\mathbb{KL}$ term (see appendix for details). For MNIST and Omniglot we used a fixed binarization and a Bernoulli decoder, whereas for FMNIST we use a Gaussian decoder. Table 1 shows test log-likelihoods (not ELBOs, these are obtained as in Burda et al. [4] with $m = 1000$, and are computed on the recovered discrete model) plus/minus one standard deviation for two different architectures. We highlight best results and those within error. The IGR performs best or is within error, except in a single scenario. We report the test log-likelihood as it is the most relevant metric from a machine learning perspective, but from an optimization point of view, the discretized training ELBO is of more interest, as it more accurately measures how well the original objective is being maximized. We include this evaluation, which is also favorable to IGR, in the appendix. It is also worth mentioning that the execution times between the IGR and the GS were almost identical for the I and SB variants. Nonetheless, the IGR Planar is about 30% slower than all the other alternatives.

To verify how much of our performance improvement is due to our closed form $\mathbb{KL}$, we also trained the VAE using the *sticking the landing* gradient estimator proposed by Roeder et al. [28], which does not involve a closed form $\mathbb{KL}$ divergence. Results are also shown in Table 1 (with the label SL). Note that all SL models outperform their non-SL counterparts, suggesting that the closed form $\mathbb{KL}$ of the IGR is not a key component of its superior empirical performance. We note that closed form $\mathbb{KL}$ remains an attractive theoretical property which could prove more useful in other applications.

In Figure 2 we show that the IGR also outperforms the GS on the continuous model (not only the discrete one). The plot contains error bars, but these are almost imperceptible due to their size and the scale of the plot. Note that while we include this comparison for completeness, as we believe that

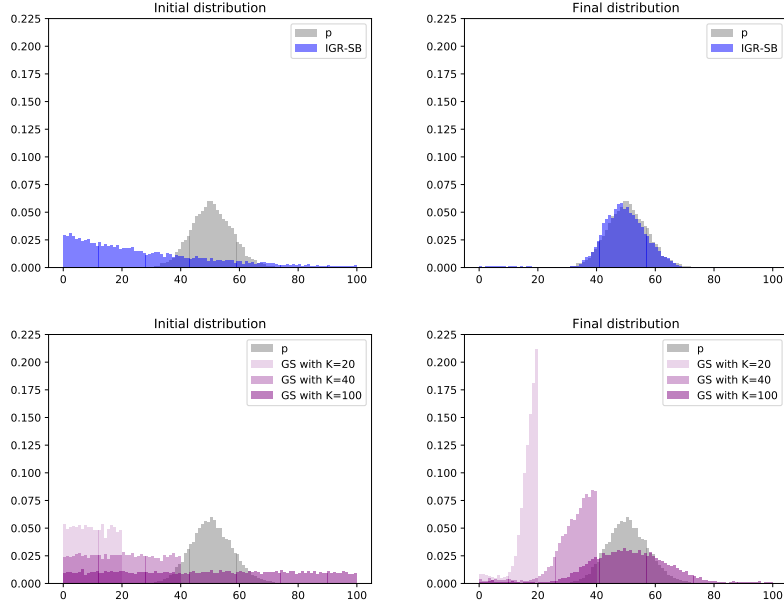

Figure 1: Approximations to a Poisson distribution for IGR-SB (top right panel) and GS (bottom right panel) after 1,000 training steps. Initial values of the approximations are displayed on the respective left panels. Due to the stick-breaking procedure, a random initialization concentrates to the left.

| Architecture | Discrete Models | MNIST | FMNIST | Omniglot |
|---|---|---|---|---|
| Linear | IGR-I | **-94.65 ± 0.14** | **-38.12 ± 0.12** | -128.14 ± 0.40 |
| | IGR-Planar | -96.21 ± 0.14 | -38.72 ± 0.17 | -130.76 ± 0.17 |
| | IGR-SB | -96.74 ± 0.36 | -41.70 ± 0.50 | **-124.77 ± 0.40** |
| | GS | -106.17 ± 1.00 | -46.65 ± 0.89 | -138.98 ± 1.01 |
| Non-linear | IGR-I | **-91.98 ± 1.29** | -34.80 ± 3.33 | **-135.30 ± 1.71** |
| | IGR-Planar | **-92.91 ± 2.51** | -34.10 ± 3.23 | **-133.63 ± 1.86** |
| | IGR-SB | -94.92 ± 0.66 | **-34.57 ± 3.09** | -139.82 ± 9.27 |
| | GS | -98.06 ± 1.73 | **-29.72 ± 2.77** | -147.71 ± 3.04 |
| Linear | IGR-I + SL | **-94.18 ± 0.37** | **-38.16 ± 0.35** | **-122.96 ± 1.32** |
| | IGR-Planar + SL | -95.97 ± 0.53 | **-38.59 ± 0.29** | -127.96 ± 3.75 |
| | IGR-SB + SL | -96.05 ± 0.74 | -39.52 ± 0.32 | **-124.35 ± 1.10** |
| | GS + SL | -103.80 ± 0.73 | -43.86 ± 1.22 | -133.45 ± 1.88 |
| Non-Linear | IGR-I + SL | -91.38 ± 0.86 | -34.39 ± 0.67 | -134.60 ± 0.68 |
| | IGR-Planar + SL | **-88.81 ± 0.49** | -33.99 ± 1.82 | **-129.47 ± 1.06** |
| | IGR-SB + SL | -92.67 ± 1.48 | -34.86 ± 1.30 | -135.82 ± 2.58 |
| | GS + SL | -97.87 ± 0.61 | **-28.81 ± 0.64** | -140.37 ± 0.25 |

Table 1: Test log-likelihood on MNIST, FMNIST and Omniglot for IGR and GS. Higher is better.

the most relevant comparisons are on the recovered discrete model, it is interesting to see that the performance gains of the IGR over the GS on discretized models do not come at the cost of poorer continuous ones.

Finally, we compared IGR and the GS using the variance reduction technique of Grathwohl et al. [10], whose use is enabled thanks to proposition 2. We include this comparison — which was yet again favorable to IGR — and the corresponding discussion, along with a comparison against the estimator proposed by Kool et al. [16], in the appendix.

## 4.3 Structured Output Prediction

We consider a Structured Output Prediction task, where we reconstruct the lower part of an image given the upper part by using a binary stochastic feedforward neural network. In contrast to our

| Discrete Models | MNIST |
|---|---|
| **IGR-I** | $-57.28 \pm 0.07$ |
| **IGR-Planar** | $-56.61 \pm 0.13$ |
| **IGR-SB** | $\mathbf{-45.12 \pm 1.61}$ |
| **GS** | $-59.31 \pm 0.21$ |

Table 2: Test log-likelihood on MNIST. Higher is better.

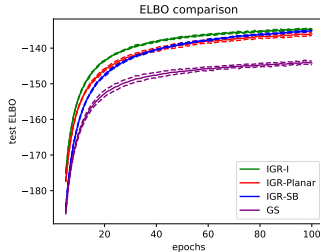

Figure 2: Test ELBO (of continuous model) on MNIST. Higher is better. This result is expected to defer from the discrete log-likelihood in Table 1 (see appendix for details).

| Discrete Models | MNIST | Omniglot |
|---|---|---|
| **IGR-SB-MM + SL** | $\mathbf{-81.79 \pm 0.35}$ | $\mathbf{-129.06 \pm 0.30}$ |
| **DLDPMM + SL** | $\mathbf{-81.90 \pm 0.30}$ | $\mathbf{-128.72 \pm 0.22}$ |

Table 3: Test log-likelihood on MNIST, FMNIST and Omniglot for IGR and GS. Higher is better.

previous experiments, this is a task that does not require the computation of a $\mathbb{KL}$ divergence. It was first proposed in Raiko et al. [24] and replicated by Gu et al. [11], Jang et al. [12] and Maddison et al. [19], and does not involve a $\mathbb{KL}$. The results of this experiment are in Table 2, where we can see that once again, IGR outperforms the GS.

### 4.4 Nonparametric Modeling

In our last experiment, we show that the truncation strategy that we use in equation 19 not only enables the use of continuous relaxations in the countably infinite setting, but that it also allows us to have reparameterizable distributions on $\mathcal{S}^\infty$ (the difference being that one should concentrate most of its mass around the vertices while the other one not necessarily). In order to show this, we follow Nalisnick et al. [23], who use a VAE with a Dirichlet Process mixture of Gaussians as the prior. For the approximate posterior, they apply the stick-breaking procedure to $K$ Kumaraswamy random variables, where $K$ has to be specified in advance and is thus treated as a hyperparameter [22]. We note that this prespecification of $K$ is problematic, as while the prior remains nonparametric, the resulting optimization objective matches the ELBO that would be obtained using a Dirichlet mixture of Gaussians with $K$ components as the prior, effectively losing the nonparametric aspect of the model. In contrast, we use an IGR distribution as the approximate posterior, and the truncation strategy from equation 19 allows us to retain the true nonparametric nature of the model.

Since this task does not require a continuous relaxation but just a reparameterizable distribution on the simplex, we use equation 17 but replace $g$ with the identity function, thus dropping the temperature hyperparameter. We use the label IGR-SB-MM to make this explicit, and use the label DLDPMM for the model of Nalisnick et al. [23]. Table 3 compares the two methods, where we trained DLDPMM with $K = 7, 9, 11, 13, 15, 17$ and report the best result. We can see that not only does IGR enable truly nonparametric inference thus not requiring an expensive hyperparameter search over $K$, but also that this does not come at the cost of decreased performance. We also note that a single IGR-SB-MM run takes the same amount of time as a single DLDPMM run.

## 5 Conclusion

In this paper we propose IGR, a flexible discrete reparameterization as an alternative to the GS in which Gaussian noise is transformed through an invertible function onto the simplex. At the cost of losing the parameter interpretability of the GS, our method results in a more natural and more flexible distribution, which has the further advantage of admitting closed form $\mathbb{KL}$ evaluation. We show that IGR significantly outperforms the GS and that, perhaps surprisingly, this improvement is not due to this nice theoretical property. Finally, IGR also extends the reparameterization trick to discrete distributions with countably infinite support and can be incorporated in nonparametric settings.

## Broader Impact

We do not foresee our work having any negative ethical implications or societal consequences.

## Acknowledgments and Disclosure of Funding

We thank Harry Braviner for useful comments and we also thank the Simons Foundation, Sloan Foundation, McKnight Endowment Fund, NIH NINDS 5R01NS100066, NSF 1707398, and the Gatsby Charitable Foundation for support.

## Footnotes

*Work partially done while at the Department of Statistics, Columbia University.

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
