[Supplementary Material]

# Appendix for: Invertible Gaussian Reparameterization

## 1 Computing the determinant of the Jacobian of the softmax$_{++}$

As mentioned in section 3.1, we can use the matrix determinant lemma to efficiently compute the determinant of the Jacobian of the softmax$_{++}$. It is straightforward to see that, for $g$ as in equation 14 from the main manuscript, we have:

$$
J_g(y,\tau) = \frac{1}{s^2}
\begin{pmatrix}
\frac{1}{\tau}y_1 e^{y_1/\tau} - e^{2y_1/\tau} & -e^{(y_1+y_2)/\tau} & \cdots & -e^{(y_1+y_{K-1})/\tau} \\
-e^{(y_2+y_1)/\tau} & \frac{1}{\tau}y_2 e^{y_2/\tau} - e^{2y_2/\tau} & \cdots & -e^{(y_2+y_{K-1})/\tau} \\
\vdots & \vdots & \ddots & \vdots \\
-e^{(y_{K-1}+y_1)/\tau} & -e^{(y_{K-1}+y_2)/\tau} & \cdots & \frac{1}{\tau}y_{K-1}e^{y_{K-1}/\tau} - e^{2y_{K-1}/\tau}
\end{pmatrix}
\tag{1}
$$

where:

$$
s = \sum_{k=1}^{K-1} e^{y_k/\tau} + \delta
\tag{2}
$$

Then, if we define $v = (e^{y_1/\tau},\ldots,e^{y_{K-1}/\tau})^\top$ and $D = \mathrm{diag}(y/\tau \odot e^{y/\tau})$, where $\odot$ represents element-wise product and $e^y$ is also taken as an element-wise operation, we have that:

$$
\det\left(J_g(y,\tau)\right) = \det\left(\frac{1}{s^2}\left(D - vv^\top\right)\right) = \left(\frac{1}{s^2}\right)^{K-1}\left(1 - v^\top D^{-1} v\right)\det D
\tag{3}
$$

$$
= \left(\frac{1}{s^2}\right)^{K-1}\left(1 - \tau \sum_{k=1}^{K-1}\frac{e^{y_k/\tau}}{y_k}\right)\frac{1}{\tau^{K-1}}\prod_{k=1}^{K-1} y_k e^{y_k/\tau}
\tag{4}
$$

where the second equality follows from the matrix determinant lemma.

## 2 Proofs of propositions

**Proposition 1**: For any $\delta > 0$, the following holds:

$$
\lim_{\tau \to 0} \mathrm{softmax}_{++}(y/\tau) = h(y) := 
\begin{cases}
e_{k^*}, & \text{if } k^* = \arg\max_{k=1,\ldots,K-1}(y_k) \text{ and } \max_{k=1,\ldots,K-1}(y_k) > 0 \\
0, & \text{if } \max_{k=1,\ldots,K-1}(y_k) < 0
\end{cases}
\tag{5}
$$

where $e_k \in \mathbb{R}^{K-1}$ is the one-hot vector with a 1 in its $k$-th coordinate.

**Proof**: We will assume that $\arg\max_{k=1,\ldots,K-1}(y_k)$ is unique, and denote $y^* = \max_{k=1,\ldots,K-1}(y_k)$. We then have:

$$
\mathrm{softmax}_{++}(y/\tau)_k = \frac{e^{y_k/\tau}}{\sum_{j=1}^{K-1} e^{y_j/\tau} + \delta} = \frac{e^{(y_k-y^*)/\tau}}{\sum_{j=1}^{K-1} e^{(y_j-y^*)/\tau} + \delta e^{-y^*/\tau}}
\tag{6}
$$

If $y_k < y^*$, the numerator goes to 0 as $\tau \to 0$, while it is equal to 1 if $y_k = y^*$. The denominator goes to $\infty$ if $y^* < 0$, while it goes to 1 if $y^* > 0$. Combining these observations finishes the proof. $\quad\square$

**Proposition 2**: If $y_k \sim \mathcal{N}(\mu_k, \sigma_k)$ for $k = 1, \ldots, K - 1$, and we define the discrete random variable $H$ by $H = k$ if $h(y) = e_k$ and $H = K$ if $h(y) = 0$, then:

$$
\mathbb{P}(H = k) = \begin{cases} \displaystyle\int_0^\infty \phi\left(\frac{t - \mu_k}{\sigma_k}\right) \prod_{j \neq k} \Phi\left(\frac{t - \mu_j}{\sigma_j}\right) dt, \text{ if } k = 1, \ldots, K - 1 \\ \displaystyle\prod_{j=1}^{K-1} \Phi\left(-\frac{\mu_j}{\sigma_j}\right) \qquad\qquad\qquad\qquad , \text{ if } k = K \end{cases}
\tag{7}
$$

where $\phi$ and $\Phi$ are the standard Gaussian pdf and cdf, respectively.

**Proof**: For $k = 1, \ldots, K - 1$, we have:

$$
\mathbb{P}(H = k) = \int_{\{y : y_k \geq y_1, \ldots, y_k \geq y_{K-1}, y_k \geq 0\}} p(y) dy
\tag{8}
$$

$$
= \int_0^\infty \int_{-\infty}^{y_k} \cdots \int_{-\infty}^{y_k} \prod_{j=1}^{K-1} \frac{1}{\sigma_j} \phi\left(\frac{y_j - \mu_j}{\sigma_j}\right) dy_1 \cdots dy_{k-1} dy_{k+1} \cdots dy_{K-1} dy_k
\tag{9}
$$

$$
= \int_0^\infty \frac{1}{\sigma_k} \phi\left(\frac{y_k - \mu_k}{\sigma_k}\right) \prod_{j \neq k} \left(\int_{-\infty}^{y_k} \frac{1}{\sigma_j} \phi\left(\frac{y_j - \mu_j}{\sigma_j}\right) dy_j\right) dy_k
\tag{10}
$$

$$
= \int_0^\infty \frac{1}{\sigma_k} \phi\left(\frac{y_k - \mu_k}{\sigma_k}\right) \prod_{j \neq k} \Phi\left(\frac{y_k - \mu_j}{\sigma_j}\right) dy_k
\tag{11}
$$

which finishes the first part of the proof. The remaining probability, $\mathbb{P}(H = K)$ can obviously be recovered as one minus the sum of the above probabilities, but we can also obtain the following expression:

$$
\mathbb{P}(H = K) = \int_{\{y : y_1 \leq 0, \ldots, y_{K-1} \leq 0\}} p(y) dy
\tag{12}
$$

$$
= \int_{-\infty}^0 \cdots \int_{-\infty}^0 \prod_{j=1}^{K-1} \frac{1}{\sigma_j} \phi\left(\frac{y_j - \mu_j}{\sigma_j}\right) dy_1 \cdots dy_{K-1}
\tag{13}
$$

$$
= \prod_{j=1}^{K-1} \Phi\left(-\frac{\mu_j}{\sigma_j}\right)
\tag{14}
$$

which finishes the proof. $\quad\square$

In our experiments with RELAX [1] in section 4 of the appendix we approximate the required integrals using a Gaussian quadrature as in Steen et al. [6], and backpropagate through this procedure. Note that the involved integrals are one-dimensional and thus can be accurately approximated with quadrature methods. Although we found better performance with these approximations than with a Monte Carlo approximation, we found the method prone to numerical instabilities, which we solved by limiting the range of values that $\mu$ and $\sigma$ are allowed take as follows:

$$
\begin{aligned} \mu &= -5 \tanh\left(\mu'\right) \\ \sigma &= 0.5 + 2\,\text{sigmoid}\left(\sigma'\right) \end{aligned}
\tag{15}
$$

where $\mu'$ and $\sigma'$ are the parameters that we optimize over.

## 3  Variational Autoencoders

As mentioned in the main manuscript, our VAE experiments closely follow Maddison et al. [4]: we use the same continuous objective and the same evaluation metrics. The experiments differ to Jang

| Model | MNIST | FMNIST | Omniglot |
|---|---|---|---|
| IGR-I | -131.86 | -66.74 | **-143.09** |
| IGR-Planar | **-126.44** | **-53.65** | -161.78 |
| IGR-SB | -130.99 | -71.87 | -154.23 |
| GS | -147.82 | -85.74 | -160.46 |
| IGR-I + SL | -128.79 | -65.77 | -145.66 |
| IGR-Planar + SL | **-126.22** | -66.39 | **-139.56** |
| IGR-SB + SL | -128.08 | **-65.21** | -157.11 |
| GS + SL | -147.60 | -83.24 | -155.17 |

Table 1: Discretized Train ELBO (not log-likelihood) on MNIST, FMNIST and Omniglot for IGR and GS. Higher is better.

et al. [2] since they use a $\mathbb{KL}$ term as in equation 8 of the main manuscript, whereas Maddison et al. [4] use a continuous $\mathbb{KL}$ as in equation 9 of the main manuscript. Using the former $\mathbb{KL}$ results in optimizing a continuous objective which is not a log-likelihood lower bound anymore, which is mainly why we followed Maddison et al. [4].

In addition to the reported comparisons in the main manuscript, we include further comparisons in Table 1 reporting the discretized training ELBO instead.

## 4 Other Estimators

Tucker et al. [7] and Grathwohl et al. [1] proposed REBAR and RELAX, respectively. These are variance reduction techniques which heavily lean on the GS to improve the variance of the obtained gradients. We make several important notes: First, REBAR is a special case of RELAX, so that we will only compare against RELAX. Second, RELAX takes advantage of the parameter interpretability of the GS, as it considers the gradients of the relaxed objective as approximations to the gradients of the objective of interest:

$$\nabla_\alpha \mathbb{E}_{z \sim \alpha}[f(z)] \approx \nabla_\alpha \mathbb{E}_{\tilde{q}_{\alpha,\tau}(\tilde{z})}[\tilde{f}(\tilde{z})] \tag{16}$$

where $\alpha$ is a discrete distribution, which we think of as a vector of length $K$ and $\tilde{q}_{\alpha,\tau}$ is a GS distribution. RELAX builds upon equation 16 to develop an estimator with reduced variance. Extending this observation to IGR is not immediately straightforward, as $\nabla_{\mu,\sigma} \mathbb{E}_{q_{\mu,\sigma,\tau}(\tilde{z})}[\tilde{f}(\tilde{z})]$ is not an approximation to the gradient on the left hand size of the above equation: it is not even the same shape. However, thanks to proposition 2 we can parameterize a discrete distribution using $\mu$ and $\sigma$, so that $\alpha(\mu,\sigma)$ is the discrete distribution given by proposition 2. This way, instead of directly optimizing over the discrete distribution, we optimize over its parameters, $\mu$ and $\sigma$, so that the gradient of interest becomes $\nabla_{\mu,\sigma} \mathbb{E}_{z \sim \alpha(\mu,\sigma)}[f(z)]$, and its corresponding approximation:

$$\nabla_{\mu,\sigma} \mathbb{E}_{z \sim \alpha(\mu,\sigma)}[f(z)] \approx \nabla_{\mu,\sigma} \mathbb{E}_{\tilde{q}_{\mu,\sigma,\tau}(\tilde{z})}[\tilde{f}(\tilde{z})] \tag{17}$$

where $\tilde{q}_{\mu,\sigma,\tau}$ is an IGR distribution, thus enabling the use of RELAX along IGR. Third, it should also be noted that the bias and variance of the gradient estimator of RELAX are central points of discussion by Grathwohl et al. [1]. However, comparing bias and variance between the GS and IGR is a difficult task, as they are intrinsically approximating different gradients (equations 16 and 17, respectively). To make the fairest possible comparison, we compare between IGR and the GS not by trying to estimate biases and variances, but by empirically comparing the recovered discrete objectives. Ultimately, bias and variance of a stochastic gradient estimator are used as proxies for how adequately optimized the corresponding objective will be, so that directly comparing on this metric is sensible. We show results of running IGR and GS with and without RELAX in Table 2.

| Discrete Models | MNIST |
|---|---|
| IGR-I | -94.18 |
| GS | -103.80 |
| IGR-I + RELAX | **-81.95** |
| GS + RELAX | -83.41 |

Table 2: Test log-likelihood on MNIST for nonlinear architecture. Higher is better.

Finally, Kool et al. [3] proposed USPGBL, an unbiased estimator (unlike the GS or IGR, which are biased), which is based on sampling without replacement. Their method requires using several approximate posterior samples to estimate the ELBO. We used $S = 4$ samples, and for a fair comparison against GS and IGR, we also estimated the ELBO using $4$ samples (instead of $1$, which we used in every other experiment). Results are in Table 3 and we can see that again, IGR performs best.

| Discrete Models | MNIST ($S = 4$) |
|---|---|
| IGR-I | -118.45 |
| GS | -126.84 |
| IGR-I + RELAX | **-102.21** |
| GS + RELAX | -112.76 |
| USPGBL | -106.89 |

Table 3: Test ELBO on MNIST for nonlinear architecture with 4 samples. Higher is better.

## 5 Architecture and hyperparameter details

In this section we describe the hyperparameters and architecture for the VAEs we use. The choice of hyperparameters and architecture are aligned with [4, 2, 7, 1, 3].

- **Linear Architecture: 784 - 200 - 784**
  - Encoder: One fully connected dense layers of 200 units with linear activation.
  - Decoder: Symmetrical to the Encoder. One fully connected dense layer of 200 units with linear activation.
- **Non-linear Architecture: 784 $\sim$ 512 $\sim$ 256 - 200 - 256 $\sim$ 512 $\sim$ 784**
  - Encoder: Two fully connected dense layers of 512 units and 256 units respectively. The nonlinear activations are ReLu.
  - Decoder: Symmetrical to the Encoder. Two fully connected dense layers of 256 units and 512 units respectively. The nonlinear activations are ReLu.

The hyperparameters are shared across the models. The only thing that changes is the temperature, which is selected through cross validation as specified in the main manuscript. We use the following configuration:

- Batch size = 100
- Epochs = 300 - 500
- Learning Rate $\in \{1.e - 4, 3.e - 4\}$
- Adam with $\beta_1 = 0.9, \beta_2 = 0.999$
- Categories = 10
- Number of Discrete Variables = 20

For the **structure output prediction** task the architecture used is:

- **Four-layered non-linear architecture 240 $\sim$ 240 - (first sample) & 240 $\sim$ 240 - (second sample)**
  - First sample is taken from double-layer 240 with tahn activation followed by a layer with 240 units with a linear activation
  - Second sample is taken as above

The hyperparameters used are

- Batch size = 100
- Epochs = 100
- Learning Rate = 1.e-3

- Weight Decay = 1.e-3
- Adam with $\beta_1 = 0.9, \beta_2 = 0.999$

For the **nonparameteric mixture model** the architecture used is:

- **Nonlinear architecture 784 $\sim$ 200 $\sim$ 200 - 200 - 200 $\sim$ 200 - 784**
    - Encoder: 3 fully connected dense layers with 200 units and a ReLu activation.
    - Decoder: 3 fully connected dense layers with 200 units and a ReLu activation.

The hyperparameters used are

- Batch size = 100
- Epochs = 300
- Learning Rate = 3.e-4
- Continuous Dimensionality = 50
- Max number of mixtures = 20 (not necessarily used)
- Adam with $\beta_1 = 0.9, \beta_2 = 0.999$

## 6 Approximating Discrete Distributions

Next we compare the GS and the IGR in approximating discrete distributions. We took 1,000 samples of the learned parameters of the IGR from solving equation 14 from the main manuscript.

Figure 1: IGR approximation to a Binomial($N = 12, p = 0.3$)

Figure 2: GS approximation to a Binomial($N = 12, p = 0.3$)

We observe how both methods approximate the Binomial adequately, although it seems that the advantage of the IGR-SB to better approximate countably infinite distributions was not translated to this simple example.

Figure 3: IGR approximation to a Discrete defined as $p = \left( \frac{10}{46}, \frac{1}{46}, \frac{5}{46}, \frac{1}{46}, \frac{10}{46}, \frac{10}{46}, \frac{1}{46}, \frac{6}{46}, \frac{1}{46}, \frac{1}{46} \right)$.

Figure 4: GS approximation to a Discrete defined as $p = \left( \frac{10}{46}, \frac{1}{46}, \frac{5}{46}, \frac{1}{46}, \frac{10}{46}, \frac{10}{46}, \frac{1}{46}, \frac{6}{46}, \frac{1}{46}, \frac{1}{46} \right)$.

Results for this discrete distribution are similar to those observed on the Binomial.

Figure 5: IGR-SB approximation to a Negative Binomial$(r = 50, p = 0.6)$.

Figure 6: IGR-SB approximation to a Negative Binomial$(r = 50, p = 0.6)$.

Here again we see how the GS has difficulty approximating another distribution with a countably infinite support. The GS with $K = 40$ (middle-purple) doest not assign mass to the right tail where as the GS with $K = 100$ has difficulty taking out sufficient weight from the right tail of the distribution.