[Reviews · NeurIPS 2020]

Review 1

Summary and Contributions: Update: Thanks for the rebuttal, which resolves some of my concerns, including some errors in my previous review. Now my biggest concern is that the novelty of the proposed method is a bit incremental given the GS, because what it essentially does is to define a more complex transformation. This can be strengthen by more experiment verifications, but was not provided in the paper. Therefore, I am not fully confident to support the paper, but I will update my score to 5. =========== This paper proposes an alternative of the popular Gumbel-Softmax trick, called invertible Gaussian reparameterization. The idea is to transform Gaussian noise to a one-hot approximation through an invertible function. Some experiments have verified the effectiveness of the proposed technique.

Strengths: The main strength is the proposal of a more flexible reparametrization trick than the Gumbel-Softmax trick, and empirical results seem to achieve better results.

Weaknesses: The writing is not good enough, resulting in a number of confusions that will be detailed below. Also, the empirical results do not seem significant enough.

Correctness: The claims and method seem to be correct. However, I think there is a significant issue in eq. 13. According to eq.12, \tilde{q} is not gaussian for sure, but how come eq.13 is true? eq.13 seems to say that \tilde{q} is gaussian, which I think is wrong.

Clarity: The paper is not well written. See more details below.

Relation to Prior Work: Yes.

Reproducibility: Yes

Additional Feedback: First, I think this paper is not well written. The flow of introduction of the proposed method is not well structured, making the paper hard to follow. For example, the standard Gumbel-Softmax basically transforms the parameter (\alpha in the paper) of a discrete distribution to an approximate sample from the discrete distribution. However, for the proposed IGR, it is not clear what this corresponds to? Is it \mu or \sigma in eq.10? There are also some other issues. For example, eq.14 seems wrong because according to previous sections, p_0 is a distribution. How is eq.14 a valid formulation with the existence of p_0? The complete words of pdf and cdf in Proposition 2 should be written out. Finally, the experiments are not convincing. First, the tasks and data used are too simple. A more complicated task with large data is expected. Second, the numbers do not seem significantly different from those of GS. Also, comparison in terms of efficiency by including the running time will be a plus.


Review 2

Summary and Contributions: This paper discusses alternatives to the Gumbel-Softmax relaxation for Categorical distributions. The proposed approach is to sample from a Gaussian and map the sample through an invertible transformation that maps from R^{K-1} to the probability simplex with K vertices. The authors discuss two such mappings: (i) an invertible softmax, where one of the logits is fixed, and (ii) through the (invertible) stick-breaking procedure, which can be adapted to the countably infinite setting. Since transformations are invertible, a KL on the final transformed sample is the same as the KL on the base Gaussian distribution, so a closed form KL is available. To add a temperature parameter, the authors either add this to the softmax in the standard way, or interpolate between the point on the simplex and its closest vertex. Finally, to recover the closest discrete distribution, the softmax can be replaced with an argmax. Post-rebuttal: Thanks for the clarifications.

Strengths: The proposed approach is simple, has nice properties, and extensible.

Weaknesses: I don’t see any major weaknesses, but there are points I’d like to see the authors comment on: The Gumbel-Softmax distribution has a very intuitive discrete limit. While the downside of lacking interpretability for the proposed IGT is discussed by the authors, I wonder if the IGT can be used with the straight-through gradient estimator? Many applications require that the samples be discrete, and in such cases, Gumbel-Softmax + Straight-through still seems to be the better solution than IGT. I wonder if the authors had done any experiments for testing if IGT can also be used with the straight-through gradient? If I understood correctly, the stick-breaking procedure requires an ordering to the K classes and necessarily places decreasing mass according to this ordering. This seems to not be discussed in the paper? Would this be a reason why SB performs poorly on the VAE task (with K=10) but does much better on the binary stochastic NN?

Correctness: Minor: the notation in eq (8) could be clarified a bit: alpha bar is a scalar, not a probability distribution. Do the authors mean to denote a Categorical distribution with probabilities equal to alpha bar?

Clarity: The paper is generally very well-written and easy to follow. There were two sentences that stood out to me: “Indeed, Stirn et al. [...] discrete distributions.” seems to be out of place. I did not understand what this has to do with the temperature hyperparameter. Perhaps this would be better discussed elsewhere as related work.

Relation to Prior Work: Related works have been discussed up front by the authors.

Reproducibility: Yes

Additional Feedback:


Review 3

Summary and Contributions: This paper introduces an alternative to the Gumbel-Softmax: a novel reparameterization for relaxed discrete distributions using invertible transformations and a Gaussian base distribution (which is compatible with the reparameterization trick). 3 transformations are considered: a modified softmax, a stick-breaking procedure and normalizing flows. Notably, the stick-breaking procedure allows modelling distribution allows applying the reparameterization to discrete distributions with infinite support. The method is empirically validated on fitting Poisson distributions (infinite support), training discrete VAEs and structured output prediction. Evaluation is performed on the equivalent discrete models.

Strengths: The authors proposed a novel approach for the reparameterization-compatible relaxation of categorical distributions, which is fundamental to the learning of discrete variable models. The experiments show significant improvements over Gumbel-Softmax (based on the final test log-likelihood of the corresponding discrete generative models and a structured output prediction task).

Weaknesses: The experimental data is not well presented. No further analysis is presented to explain why the results are superior. [Update] Thank you for your feedback. (1) I agree with Reviewer 4 that focusing on linear models is not sufficient, some results are reported in the appendix however they are not entirely convincing (FMNIST: such a large variance in results is unexpected). The performances of relaxation-based methods greatly depend on the choice of the relaxation parameter (temperature) and choosing the right temperature becomes increasingly difficult as the complexity of the model increases. Hence, discussing the effect of the temperature parameter, studying deeper models and comparing the methods with relaxation-free methods is necessary in my opinion. (2) Clarity of the story line: (a) because of point (1), the empirical results do lack of strength. (b) The paper introduces 3 sub-methods (I, Planar, SB) (+3 "KL" expressions for GS which collapse to 1 in the experiments) and experiments are briefly presented without further interpretation (c) it is unclear what "conceptually simpler" means. The probabilistic nature of the Concrete/GS distribution is indeed not obvious, yet the "softmax + gumbel noise" foundation is appealing. Whereas I believe your idea is sound, I won't update my grading due to the 2 points mentioned above.

Correctness: Theory: The theoretical claims seem correct (based on following the proofs in the appendix). Results: Experiments should be run using the existing benchmarks and should aim at reproducing (part of) the tables stated in previous work, otherwise, it is difficult to trust the empirical results. In the appendix, table 2, IGR-I + RELAX scores -70.41 nats on a binarized MNIST dataset, which slightly differs from the standard statistically binarized MNIST dataset from [1], for which the state-of-the-art is 78.01 [2]. This result could be correct (since the datasets differ) but require further confirmation from the author -- in the best scenarios, experiments should rely on the original binarized MNIST dataset. [1] Salakhutdinov, Ruslan, and Iain Murray. "On the quantitative analysis of deep belief networks." Proceedings of the 25th international conference on Machine learning. 2008. [2] Vahdat, Arash and Jan Kautz. “NVAE: A Deep Hierarchical Variational Autoencoder.” (2020). [update] thank you for the clarification on that point.

Clarity: The storyline is somewhat confusing as it seems to lack of a clear storyline. The experiments lack interpretation.

Relation to Prior Work: * citation [17], line 19: This works aims at reducing the variance of the pathwise gradient estimator, not the score function estimator. * citation [22], line 257: Indeed, a Monte-Carlo estimate of the KL is used in Sticking the Landing, however, this is not a contribution of the paper. In fact, using a Monte-Carlo estimate of the KL is commonplace when training VAEs.

Reproducibility: Yes

Additional Feedback: Experimental protocol: All results are reported on the test set, optimization performances should be measured on the training set in order to avoid measuring overfitting. The paper focuses on the close form expression of the KL (which is important in essence), whereas it is of little importance for the considered experiments. Overall the idea has a high potential and the experiments are promising. However, the writing and the references to the related work show a slight lack of maturity.


Review 4

Summary and Contributions: This paper proposes a new reparameterizable distribution over the probability simplex that can be used as an alternative to discrete distributions and is amenable to backpropagation, in the same vein as Gumbel-Softmax (GS) / Concrete distribution. The main construction proposed in the paper, the Invertible Gaussian Reparameterization (IGR), works by transforming a standard multinomial Gaussian through a smooth invertible function to represent a distribution on the probability simplex. This, with the addition of a temperature parameter, can be used to approximate a discrete distribution over the vertices of the said simplex. One key advantage of IGR is that it can represent distributions on countably infinite support. The paper presents empirical evidence that shows that IGR leads to better generative modeling performance on various tasks. ----- Post-rebuttal: I have read the author feedback and chose to keep my current score.

Strengths: * Novelty: The paper combines several ideas in a novel way. First, IGR takes the approach of transforming a simple distribution via an invertible mapping, which allows the algorithm designer to have different options for the mapping. The experimental results include comparisons of different invertible mappings. Second, IGR uses the stick-breaking procedure, which allows it to represent discrete distributions with countably infinite support. These aspects make IGR substantially different from existing work. * Theoretical grounding: Propositions 1 establishes an important theoretical property of IGR as a replacement to GS, as it allows the user to recover the discrete distribution from the relaxation represented by IGR. Also the authors convincingly motivate the reason behind their design choices (e.g . the use of Gaussian as the base distribution, use of invertible transformation, and the need for stick-breaking procedure as well as softmax++) in Section 3.

Weaknesses: * Experiments: While the proposed method outperformed existing approaches in the results presented, the experiments seem rather limited in scope. For example, the VAE experiments included in the main text were done using linear encoder/decoder, which is very rarely used in practice. This is particularly concerning because for the nonlinear experiment included in the appendix, GS outperformed IGR on FMNIST (see Table 1). Although this was the only dataset on which IGR didn't outperform GS, it does raise the question of how IGR would scales to more difficult tasks (e.g. training a VAE with autoregressive decoder on CIFAR10). In this light, it would really strengthen this paper if the authors could demonstrate that IGR outperforms GS on more challenging tasks as well as compared to other methods such as VIMCO [1] and VQ-VAE [2]. * Reasoning behind the claims - The paper claims that because IGR uses (2K-2) parameters (compared to K parameters for Gumbel-Softmax), it is "more flexible." But there does not seem to be any justification for this. Since IGR and GS use different parameterizations, it'd be be informative to have some evidence (either theoretical or empirical) behind this claim. For example, how does the performance gap between IGR and GS change for different values of K? - The paper mentions that IGR is "conceptually simpler" than GS. However, the authors did not convincingly demonstrate that their method is conceptually simpler, nor did their experiments show that their new method had any practical benefits. * Related work: This paper is missing a related work section. While it's not absolutely necessary to have a dedicated section on related work, it is important to delineate the proposed method from existing work. The paper essentially only discusses GS/Concrete distribution in detail, but makes no reference to e.g. VIMCO [1]. [1] Mnih, Andriy, and Danilo J. Rezende. "Variational inference for monte carlo objectives." arXiv preprint arXiv:1602.06725 (2016). [2] Van Den Oord, Aaron, and Oriol Vinyals. "Neural discrete representation learning." Advances in Neural Information Processing Systems. 2017.

Correctness: Yes. The experiments were performed in a fair manner, and the theoretical claims seem to be valid.

Clarity: Aside from minor typos and grammatical errors, the paper is well written.

Relation to Prior Work: As mentioned above, the paper could be improved by discussing existing work in more detail, especially in comparison to IGR. This would also help emphasize the novelty of the proposed approach as well.

Reproducibility: Yes

Additional Feedback:

[Author Response · NeurIPS 2020]

**Rebuttal for "Invertible Gaussian Reparameterization" (ID 10306)**: We thank the reviewers for their time and
valuable feedback. We will incorporate all minor points in the manuscript, and answer the major ones below:
• A few key misconceptions we want to strongly rebut (R1):
− We disagree that our empirical improvements are not significant. For example, REBAR obtains a $1.65\%$ relative
improvement over the GS, and RELAX $0.22\%$ over REBAR; while we obtain $6.2\%$ over the GS.
− There is no error in eq.13. While $q$ is not Gaussian, the $\mathbb{KL}$ between $q$'s is equivalent to a $\mathbb{KL}$ between Gaussians.
The $\mathbb{KL}$ is invariant to invertible transformations. We introduced the softmax$_{++}$ precisely to use this fact.
− We note that the GS does not transform $\alpha$ into an approximate one-hot sample as claimed by R1. Instead, it
transforms Gumbel noise into the sample, and the transformation depends on $\alpha$. Similarly, IGR transforms Gaussian
noise into the sample, and the transformation depends on $(\mu, \sigma)$.
• Tasks being too simple (R1 and R4): The tasks we use are the de-facto standard for benchmarking continuous
relaxations of discrete distributions in the literature, see [8, 9, 10, 16, 22, 25] from the main manuscript and [3] from
the appendix to reference a few papers whose experiments we are consistent with. Note also that while the VAE
problems we consider involve (albeit partially factorizable) $10^{20}$-dimensional discrete distributions, the main goal of
our experiments is to compare IGR and GS, not to obtain state-of-the-art VAE performance. That being said, we will
include experiments on CIFAR-10 with more complex architectures by publication time.
• Reporting results on test set (R3): Thank you, this is an important point to discuss. From an ML standpoint, we
believe test scores on discretized models are the most relevant metric. An ML practitioner wanting to train a discrete
VAE (or another ML model) will care about the recovered discrete model's performance; regardless of the optimization
objective's value. We agree that training losses (on the discrete objective) are also interesting to look into though, and
so we have included them in the table below. Note that IGR also outperforms the GS here.
• On flexibility (R3): To show that at least to a certain degree our empirical improvements do come from added
flexibility, we re-ran our experiments while fixing $\sigma = 1$ and only learning $\mu$, resulting in the same number of degrees
of freedom as the GS. While this version of IGR still outperformed the GS, it was by a much smaller margin: test log
likelihoods were $-102.81$ and $-43.83$ for MNIST and FMNIST, respectively, and $-140.37$ for Omniglot, where the GS
slightly outperformed (compare with Table 1 from the main manuscript).
• Unclear storyline (R3): We believe the conceptual simplicity of not having to involve the arguably exotic and not
extensible Gumbel distribution, along with more straightforward density derivation, closed-form $\mathbb{KL}$, and strong
empirical results do form a coherent storyline. We will convey this in a clearer way in the manuscript.
• Straight-through estimator (R2): This is a great point. Proposition 1 enables discretizing the sample, so that we can
carry out a straight-though gradient estimate. We will also include an experiment with this estimator by publication.
• Using statically binarized MNIST (R3): We are actually using this version of MNIST. The odd numbers in the
appendix table are a consequence of numerical instabilities for the quadrature we used to estimate the integral in
proposition 2, which we have corrected since submission. This issue only affects the IGR-I+RELAX entries of tables 2
and 3 in the appendix, which should be $-81.9481$ and $-102.214$, respectively. Note also that IGR still outperforms.
• On VIMCO and VQ-VAE (R4): Thank you for mentioning these papers, we will reference them. However, we
do point out that VIMCO is outperformed by USPGBL ([3] from the appendix), against which we do compare and
outperform in the appendix; and that VQ-VAE does not endow the latent variables with a "true" distribution as they use
point masses and is different to IGR and the GS in this regard.
• Stick-breaking and orderings (R2): Note that IGR-SB need not place more mass around the first vertices of the
simplex, as by choosing $(\mu, \sigma)$ appropriately mass can be shifted towards the right (e.g our Poisson experiments).
• R1: Equation 14 is valid, while $p_0$ is a distribution, we are thinking of it as a probability vector. We will further
clarify this in the manuscript.
• R1: We also point out that we did mention that IGR and the GS have the same running times (lines 253-255).
• R2: Note that $\bar{\alpha}$ is a probability vector, not a scalar; and is thus treated as a distribution in eq. 8.
46

| Model | MNIST | FMNIST | Omniglot |
|---|---|---|---|
| IGR-I | -131.86 | -66.74 | **-143.09** |
| IGR-Planar | **-126.44** | **-53.65** | -161.78 |
| IGR-SB | -130.99 | -71.87 | -154.23 |
| GS | -147.82 | -85.74 | -160.46 |
| IGR-I + SL | -128.79 | -65.77 | -145.66 |
| IGR-Planar + SL | **-126.22** | -66.39 | **-139.56** |
| IGR-SB + SL | -128.08 | **-65.21** | -157.11 |
| GS + SL | -147.60 | -83.24 | -155.17 |

Table 1: Discretized Train ELBO (not log-likelihood) on MNIST, FMNIST and Omniglot for IGR and GS. Higher is better.

[Meta-Review · NeurIPS 2020]

This paper presents a simple alternative to the Gumbel-Softmax based on Gaussians and invertible transformations to the hypersimplex. As one reviewer noted, "the proposed approach is simple, has nice properties, and extensible". Many reviewers criticized the lack of experiments on non-linear models in the main text. Some reviewers felt that the clarity of the draft could be improved, in particular the motivation. This was a borderline paper, however I would like to recommend acceptance. I found this paper to be well-written. The experiments are well-considered, careful, and show a clear benefit of the method in many settings. I agree with the reviewers that the results on non-linear models should be moved into the main text from the supplementary, and I agree more motivation for the specific choices made would be useful. My recommendation is ultimately compelled by a very important point that is demonstrated in this paper and largely ignored in the literature on relaxed gradient estimators: the Gumbel-Softmax is only special because its discrete limit is the Gibbs distribution, but this doesn't mean it's the most useful relaxation in practical deep learning applications. Please address the clarity concerns and incorporate the non-linear model results in the main text.